# Incident prolonged QT interval in midlife and late-life cognitive performance

**Claudia K. Suemoto**[1]*, **Laura E. Gibbons**[2], **Evan L. Thacker**[3], **Jonathan D. Jackson**[4], **Claudia L. Satizabal**[5,6], **Brianne M. Bettcher**[7], **Lenore Launer**[8], **Caroline Phillips**[8], **Lon R. White**[9], **Melinda C. Power**[10]

**1** Division of Geriatrics, University of São Paulo Medical School, São Paulo, Brazil, **2** Harborview Medical Center, University of Washington, Seattle, Washington, United States of America, **3** Department of Public Health, Brigham Young University, Provo, Utah, United States of America, **4** Department of Neurology, Massachusetts General Hospital, Harvard Medical School, Boston, Massachusetts, United States of America, **5** Department of Neurology, Boston University School of Medicine, Boston, Massachusetts, United States of America, **6** UT Health San Antonio, San Antonio, Texas, United States of America, **7** Departments of Neurosurgery and Neurology, Rocky Mountain Alzheimer's Disease Center, University of Colorado Anschutz School of Medicine, Aurora, Colorado, United States of America, **8** National Institute on Aging, Bethesda, Maryland, United States of America, **9** Departments of Medicine and Geriatrics, University of Hawaii John A. Burns School of Medicine, Honolulu, Hawaii, United States of America, **10** Department of Epidemiology and Biostatistics, George Washington University Milken Institute of Public Health, Washington, DC, United States of America

* cksuemoto@usp.br

## Abstract

### Background

Measures of cardiac ventricular electrophysiology have been associated with cognitive performance in cross-sectional studies. We sought to evaluate the association of worsening ventricular repolarization in midlife, as measured by incident prolonged QT interval, with cognitive decline in late life.

### Methods

Midlife QT interval was assessed by electrocardiography during three study visits from 1965/68 to 1971/74 in a cohort of Japanese American men aged 46–68 at Exam 1 from the Honolulu Heart Study. We defined incident prolonged QT as the QT interval in the upper quartile at Exam 2 or 3 after QT interval in lower three quartiles at Exam 1. Cognitive performance was assessed at least once using the Cognitive Abilities Screening Instrument (CASI), scored using item response theory (CASI-IRT), during four subsequent visits from 1991/93 to 1999/2000 among 2,511 of the 4,737 men in the Honolulu-Asia Aging Study otherwise eligible for inclusion in analyses. We used marginal structural modeling to determine the association of incident prolonged QT with cognitive decline, using weighting to account for confounding and attrition.

### Results

Incident prolonged QT interval in midlife was not associated with late-life CASI-IRT at cognitive baseline (estimated difference in CASI-IRT: 0.04; 95% CI: -0.28, 0.35; p = 0.81), or

**Data Availability Statement:** The data underlying this study is owned by the Kuankini Medical Center in Honolulu and cannot be shared publicly. Alternatively, data may be examined through

collaboration with intramural researchers, such as co-author Dr. Lenore Launer (launerl@nia.nih.gov), at the Laboratory of Epidemiology, Demography, and Biometry at the NIA where copies of HHP and HAAS data remain archived for ongoing analyses. The statistical code is available at https://github.com/gibbonsl/Code-for-Incident-prolonged-QT-interval-in-midlife-and-late-life-cognitive-performance under the MIT license.

**Funding:** These analyses were conceived and begun by a working group at the 2016 Conference on Advanced Psychometric Methods in Cognitive Aging funded by the NIH (R13 AG030995) to Dan Mungus (PI). JDJ was supported by P01 AG036694-08S1 to Sperling, LEG was supported by P50 AG05136 to Grabowski from NIH.

**Competing interests:** The authors have declared that no competing interests exist.

change in CASI-IRT over time (estimated difference in annual change in CASI-IRT: -0.002; 95%CI: -0.013, 0.010; p = 0.79). Findings were consistent across sensitivity analyses.

## Conclusions

Although many midlife cardiovascular risk factors and cardiac structure and function measures are associated with late-life cognitive decline, incident prolonged QT interval in midlife was not associated with late-life cognitive performance or cognitive decline.

## Introduction

Dementia is the sixth-leading cause of death and a major leading cause of disability in the United States. [1, 2] It is the only disease among the top 10 causes of death that has no effective treatment or prevention.[3] Over the past 20 years, research has linked several cardiovascular risk factors (CVRF) to higher risks of dementia, Alzheimer's disease, and vascular dementia. [4–7] For many CVRF, the risk for later cognitive decline depends on the timing of risk factor evaluation. Numerous studies indicate that dementia and cognitive dysfunction are more strongly associated with CVRF measured in midlife[6, 8, 9] than with CVRF assessed later in life.[10–13]

Better understanding of cardiac function in midlife may offer novel insights for dementia prevention. The QT interval on electrocardiogram represents the length of time required for the process of ventricular depolarization and repolarization, with longer QT interval indicating slower repolarization. Although clinically relevant prolonged QT interval is a known risk factor for major cardiovascular events like stroke,[14, 15] few studies have examined the potential link between ventricular repolarization and cognitive outcomes. All studies focused on cross-sectional associations with mild cognitive impairment (MCI) or overt dementia.[16–19] For example, QT dispersion, a measure of short-term variability in QT interval, was associated with worse cognitive performance in patients with MCI in small cross-sectional studies comparing individuals with normal cognition, MCI, and dementia.[16, 18] Conversely, QT interval was not cross-sectionally associated with cognitive performance in 839 very-old adults from the Chicago Health and Aging Project (mean age of 81years).[20]

We are unaware of any longitudinal studies of the association between prolonged QT interval and cognition. Despite the evidence that midlife CVRFs are often most relevant to late-life cognitive health, the longitudinal association of ventricular repolarization at midlife and late-life cognitive decline has not yet been considered. Therefore, we aimed to examine whether prolonged QT interval in midlife was predictive of late-life cognitive decline over 25 years of follow-up.

## Material and methods

### Participants

The Honolulu-Asia Aging Study (HAAS) began in 1991 and has been described in detail previously.[21] HAAS extends the Honolulu Heart Program (HHP), a prospective study of heart disease and stroke in Japanese-American men born between 1900 and 1919, who lived on Oahu in 1965. Exams 1 (1965/68), 2 (1967/70), and 3 (1971/74) were conducted in midlife as a part of the original HHP study. Beginning twenty years after Exam 3, Exams 4 (1991/93), 5 (1994/96), 6 (1997/99), and 7 (1999/2000) were conducted in late life as part of HAAS. Of the

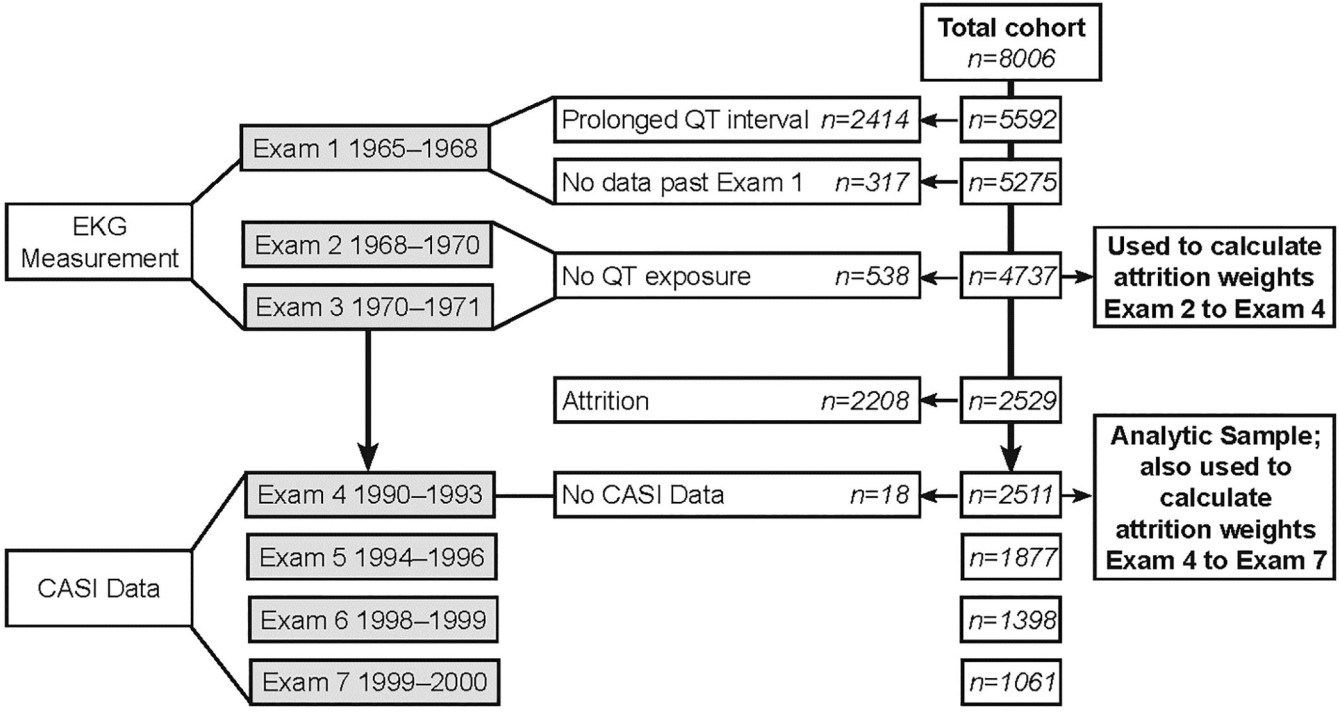

**Fig 1. Flowchart of the study participants from the Honolulu-Asia Aging study.**

original HHP cohort of 8,006 men, 2,511were enrolled in HAAS and assessed longitudinally for cognitive performance beginning at Exam 4 in 1991/93 (Fig 1). HAAS was approved by the Kuakini Medical Center Institutional Review Board and by the Human Research committees, and participants were informed about the study and signed informed consent forms.

## QT interval measurements, adjustment for ventricular rate, and identification of incident prolonged QT interval

The QT interval, which is comprised of the QRS complex, the ST segment, and T wave, is a widely-used measure of ventricular repolarization.[22] We used a correction formula that considers QT interval variation by ventricular rate.[23, 24] Specifically, we used ECG data from Exam 1 to estimate the coefficient $ß_1$ in the following linear regression model: $E$(QT interval) = $ß_0 + ß_1$(RR interval), where the RR interval is the average time elapsed in seconds between ventricular beats (60 / ventricular rate). We then used the estimated coefficient ($ß_1 = 0.158$) to calculate a ventricular rate-corrected QT interval: $QT_{adj} = QT + 0.158(1\text{-}RR)$. We also explored linear regression models in which the $ß_1$ coefficient was adjusted for age or allowed to vary with age, but found these enhancements did not meaningfully change the coefficient.

As our exposure of interest, we identified incident prolonged QT interval at Exam 2 or 3, rather than prevalent prolonged QT interval at Exam 1. Incident prolonged QT interval represents a change from faster to slower ventricular repolarization over the course of a few years, suggesting a person may be transitioning away from their normal ventricular function. Also, identifying incident prolonged QT interval at Exam 2 or 3 as our exposure of interest allowed us to use variables measured at Exam 1 to control for confounding via inverse probability weighting in the statistical analysis (see below). To identify participants who experienced incident prolonged QT interval, we first identified the 75[th] percentile of rate-corrected QT interval

at Exam 1, which was 407 milliseconds. Next, we excluded participants whose rate-corrected QT interval was above this value at Exam 1, considering them to have prevalent prolonged QT interval at Exam 1, and thus be ineligible for developing incident prolonged QT interval. Among the remaining participants, we then identified those whose rate-corrected QT interval was above 407 milliseconds at Exam 2 or 3 as having incident prolonged QT interval (Fig 1).

## Cognitive Abilities Screening Instrument (CASI)

CASI is a 40-item test of global cognitive function with scores ranging from 0 to 100. Higher scores indicate better cognitive performance.[25, 26] Inventory items were designed to be comparable to the Hasegawa Dementia Screening Scale,[27] the Mini Mental State Examination (MMSE),[28] and the Modified MMSE.[29] One additional item indexing judgment ability was also included in the inventory. To ensure equivalence across English and Japanese languages, the CASI was developed in parallel in English and Japanese at three workshops in which items were scrutinized for cultural equivalence, back-translated, and pilot-tested. CASI scores obtained in HAAS and in a purely Japanese sample were analyzed using item-response theory to further validate the cross-cultural sensitivity of CASI; no evidence of salient differential item functioning due to language of testing was found.[25] For analysis, the CASI was scored using item response theory, to measure change over time on a consistent metric;[30] the mean CASI-IRT score across Exams 4–7 was approximately normal, with mean 0 and standard deviation of 1.

## Statistical analysis

We estimated the association of incident prolonged QT interval at Exam 2 or 3 with trajectories of cognitive function from Exam 4 to Exam 7 using a marginal structural model.[31] This model uses weighting to address potential bias due to confounding and additionally implements weighting to address the potential for informative missingness related to loss to follow-up.[31, 32] To address confounding, we reweight persons in the observed sample using inverse probability of exposure weights (IPEW) to eliminate associations between potential confounders and incident prolonged QT. Thus, under standard assumptions, IPEW addresses confounding by making the exposure statistically independent of the confounders. Similarly, to reduce bias due to selective attrition, we reweight persons in the observed sample who are similar to those who drop out of the sample using inverse probability of attrition weights (IPAW). Thus, under standard assumptions, IPAW addresses informative censoring by making attrition independent of predictors of attrition, including ascertained exposure and outcome.

To establish our sample for analysis of midlife incident prolonged QT interval in relation late life to cognitive decline, we first excluded participants who did not have an electrocardiogram (ECG) at Exam 1 (n = 570), had prevalent prolonged QT interval at Exam 1 (n = 1,844), were lost to follow-up after Exam 1 (n = 317), or did not have ECG at Exam 2 or 3 (n = 538), leaving 4,737 participants (Fig 1). Data from these 4,737 persons were used to derive IPEW to account for confounding and IPAW to account for attrition from Exam 2 to Exam 4. After calculating those weights, we then further excluded participants who died or were lost to follow-up between Exam 2 and Exam 4 (n = 2,208) or had no CASI data at Exam 4 (n = 18), leaving 2,511 participants who completed CASI at Exam 4. Data from these 2,511 persons were used to derive IPAW to account for attrition from Exam 4 to Exam 7 (Fig 1).

We estimated stabilized IPEW using logistic regression models on a dataset including one line for each participant at risk of experiencing incident prolonged QT interval at either Exam 2 or Exam 3. Covariates included in the denominator model were potential confounders of the association between prolonged QT interval and cognition (S1 Table). Covariates included in

all numerator models included the subset of baseline or time-invariant covariates included in the denominator model. Stabilized IPEW were derived for each participant at each time point using standard formulae, and IPEW at each visit for each participant were then multiplied together to provide a final stabilized IPEW for each participant.[31, 32]

Similarly, we estimated stabilized IPAW to address attrition from Exam 2 through Exam 7 using logistic regression models. Because cognitive data from CASI (our outcome measure) was available only from Exam 4 onward, we modeled attrition from Exam 2 to 4 and from Exam 4 to 7 separately. Furthermore, given the expectation of two separate attrition processes—attrition due to death and attrition due to non-death drop-out—we also used separate models to account for these two processes, with models for attrition due to non-death drop out conditional on remaining alive.[33, 34]. Details of variables included in each model are available in S1 Table. As with the IPEW models, time-invariant and baseline covariates were included in the numerator models used to derive stabilized IPAW.

Stabilized weights for each process were derived for each participant at each time point, and were then multiplied together to provide a final set of stabilized IPAW or IPEW weights, with unique weights assigned to each participant at each time point. Finally, our stabilized IPEW and IPAW were multiplied together to create a unique weight for each participant contributing to the final analysis at each time point from Exams 4 to 7. Extreme weights were truncated at the 99th and 1st percentiles for use in primary analyses, as use of truncation often provides a good balance between bias and efficiency.[35]

For our primary analysis (Model 1), we estimated the association of incident prolonged QT interval on cognitive trajectories using weighted linear regression models estimated using generalized estimating equations with an independence covariance matrix.[36] We adjusted this analysis for covariates included in the models to estimate the numerators of the stabilized IPAW or IPEW, as well as important predictors of cognition: baseline age, time in study, an age by time in study interaction, generation, the presence of any *APOEε 4* alleles, education, occupation (Exam 1), height (Exam 2), chest depth (Exam 1), alcohol use (Exam 1), physical activity level (Exam 1), and history of hypertension (Exam 2). In secondary analyses, we considered the sensitivity of our findings to our modeling choices. Specifically, we compared our primary analyses to analyses using non-truncated stabilized weights (Model 2), or only stabilized IPEW weights (Model 3), complete omission of weighting, i.e., adjustment only for baseline/time-invariant covariates (Model 4), and omission of multiple imputation, i.e., analyzing only participants who had complete data on all covariates (Model 5).

Missing data in covariates used to create the weights requires either censoring of persons at the time of first missing data or implementation of methods to address missingness. Given that many participants were missing data on at least one of the covariates included in models used to create our weights, we implemented multiple imputation by chained equations (MICE) to address missing covariate data.[37] We imputed five replicate datasets after a burn-in of 10 iterations. Derivation of the weights and estimation of the MSM occurred separately within each of our five imputed datasets. Reported findings combined estimates from each of the five imputed datasets using standard methods.[38]

## Results

Of the 4,737 participants at Exam 2 (midlife), 2,511 had follow-up at Exam 4 (late life), when cognitive performance was first evaluated (Fig 1). These 2,511 were our analytic sample, weighted to represent the 4,737 who met our eligibility criteria. Their mean IRT-CASI score at Exam 4 was 0.4±0.9. Their mean corrected QT interval had been 395.1±19.3 milliseconds (range 318–475) at Exam 2, and 1,076 participants (42.9%) had incident prolonged QT interval

**Table 1. Participant characteristics at Exam 2, stratified by incident prolonged QT interval at midlife (Exam 2 or 3).**

| | Total (n = 2,511) | Prolonged QT (n = 1,076) | No Prolonged QT (n = 1,435) | p-value[a] |
|---|---|---|---|---|
| Age (years), mean (SD) | 54.5 (4.4) | 54.7 (4.6) | 54.3 (4.4) | 0.053 |
| Generation, n (%) | | | | 0.691 |
| Issei | 149 (6%) | 68 (6%) | 81 (6%) | |
| Kibei | 226 (9%) | 93 (9%) | 133 (9%) | |
| Nisei | 2,136 (85%) | 915 (85%) | 1,221 (85%) | |
| Education, n (%) | | | | 0.217 |
| None or primary | 449 (18%) | 210 (20%) | 239 (17%) | |
| Intermediate | 682 (27%) | 295 (27%) | 387 (27%) | |
| High School | 821 (33%) | 332 (31%) | 489 (34%) | |
| Technical School | 278 (11%) | 104 (10%) | 174 (12%) | |
| University | 281 (11%) | 135 (13%) | 146 (10%) | |
| Clerical, sales, professional or managerial job[b], n (%) | 787 (31%) | 368 (34%) | 419 (29%) | 0.007 |
| Hypertension diagnosis, n (%) | 225 (9%) | 107 (10%) | 118 (8%) | 0.135 |
| Alcohol (ounces/ month), mean (SD) | 11.9 (20.3) | 12.0 (19.4) | 11.8 (20.9) | 0.797 |
| Height (cm), mean (SD) | 164 (6) | 164 (6) | 163 (6) | 0.001 |
| Chest depth (cm)[b], mean (SD) | 19.2 (1.8) | 19.3 (1.9) | 19.1 (1.8) | 0.123 |
| Physical Activity Index (midlife)[b], mean (SD) | 32.9 (4.7) | 32.8 (4.6) | 33.0 (4.8) | 0.169 |
| Presence of at least one APOE-4 allele, n (%) | 462 (19%) | 218 (21%) | 244 (18%) | 0.038 |

[a] T-tests for continuous variables, chi-squared for categorical variables, and Wilcoxon's rank sum test for education.

[b] Evaluated at Exam 1

at either Exam 2 or 3. Participant characteristics at Exam 2 by incident prolonged QT interval at Exam 2 or 3 are shown in Table 1. The mean of the final stabilized weights applied to the Visit 4 data from the 2,511 participants in our analytic sample was 0.99 (range: 0.46, 1.86). Additional details about the weights are provided in the Supplemental Appendix (S2, S3, and S4 Tables).

Prolonged QT interval in midlife (Exam 2 or 3) was not associated with IRT-CASI score at Exam 4 (20 years after Exam 3) or with subsequent decline in IRT-CASI score over subsequent exams spanning the next 10 years (Table 2). In our primary analysis (Model 1), assuming reference level for all covariates, participants without midlife prolonged QT interval declined on IRT-CASI score by an average of 0.09 points per year (95% CI: 0.08 to 0.09 points per year). Participants with midlife prolonged QT interval were not significantly different in the magnitude of subsequent decline in IRT-CASI score over time (estimated difference of -0.002 points of decline per year; 95% CI: -0.013 to 0.010 points; P = 0.79). In a series of sensitivity analyses in which we applied different modeling strategies (Table 2, Models 2–5), we obtained results very similar to those from our primary analysis.

## Discussion

Using marginal structural models to reduce bias from confounding and participant attrition, we found that midlife QT interval was not associated with late-life CASI score approximately 25 years later, nor with cognitive decline in CASI over time in a large sample of Japanese American men from the HAAS. We confirmed this result in sensitivity analyses.

Certain aspects of left ventricular (LV) dimensions and ejection fraction have previously been associated with cognitive impairment and decline.[17, 39] Increased LV dimensions in 211 men who were 68 years old at baseline were associated with higher risk of cognitive decline

**Table 2. Association of elevated QT interval in midlife with item response theory-adjusted Cognitive Abilities Screening Instrument (IRT-CASI) score later in life.**

| Model and parameter | Estimated IRT-CASI score | 95% CI | P value |
|---|---|---|---|
| **Model 1** | | | |
| Study time (years) | -0.09 | (-0.09, -0.08) | <0.0001 |
| Elevated QT | 0.04 | (-0.28, 0.35) | 0.81 |
| Elevated QT × study time | -0.002 | (-0.013, 0.010) | 0.79 |
| **Model 2** | | | |
| Study time, y | -0.10 | (-0.11, -0.09) | <0.0001 |
| Elevated QT | -0.06 | (-0.46, 0.34) | 0.76 |
| Elevated QT × study time | 0.003 | (-0.013, 0.018) | 0.71 |
| **Model 3** | | | |
| Study time, y | -0.07 | (-0.08, -0.06) | <0.0001 |
| Elevated QT | 0.06 | (-0.25, 0.39) | 0.68 |
| Elevated QT × study time | -0.003 | (-0.015, 0.009) | 0.62 |
| **Model 4** | | | |
| Study time, y | -0.07 | (-0.08, -0.06) | <0.0001 |
| Elevated QT | 0.05 | (-0.27, 0.36) | 0.77 |
| Elevated QT × study time | -0.002 | (-0.014, 0.009) | 0.69 |
| **Model 5** | | | |
| Study time, y | -0.07 | (-0.08, -0.06) | <0.0001 |
| Elevated QT | 0.04 | (-0.26, 0.34) | 0.78 |
| Elevated QT × study time | -0.002 | (-0.014, 0.010) | 0.72 |

Model 1: Inverse probability of exposure weighting (IPEW) and inverse probability of attrition weighting (IPAW), with weights truncated at 1st and 99th percentiles, with multiple imputation.

Model 2: IPEW and IPAW, with weights not truncated, with multiple imputation.

Model 3: IPEW (no IPAW), with weights not truncated, with multiple imputation.

Model 4: Unweighted, with multiple imputation.

Model 5: Unweighted, without multiple imputation.

All models additionally adjusted for generation, alcohol use, physical activity level, education, occupation, and chest depth at Visit 1; age, height, and hypertension at Visit 2; and the presence of any APO-E4 alleles at visit 4.

The reference person was, at exam 2, 55 years old, 164 cm tall, and without a hypertension diagnosis. At exam 1 he was Nisei, with a primary education or less, had a chest depth of 19 cm, did not have a Clerical, sales, professional or managerial job, did not drink, and had a Physical Activity Index of 33. He also had no APOE-4 alleles.

14 years later.[17] Similarly, in 1,114 participants of the Framingham Heart Study Offspring Cohort, LV ejection fraction, an indicator of cardiac dysfunction, was associated with worse performance on neuropsychological tests related to visuospatial memory, object recognition, and executive function.[39] In addition, patients with severe LV dysfunction showed improved performance on executive and visuospatial function tests three to six months after cardiac resynchronization therapy.[40, 41] Alterations in LV structure and function may be related to cognitive impairment through either brain hypoperfusion. An additional mechanism could be the presence of ventricular arrhythmias and the generation of thromboembolism, which could lead to cerebral infarcts and transient hypoperfusion.[42]

However, the association between ventricular arrhythmias and cognitive performance has been far less investigated. Silent myocardial ischemia and repeated ventricular premature beats were more frequent in patients with MCI and Alzheimer's disease than in participants with normal cognition.[19] In a same sample of 33 patients with Alzheimer's disease, 39 with MCI, and 29 controls, QT dispersion was associated with worse performance in the MMSE.[18] In a

large sample of patients with normal LV ejection fraction, Coppola *et al.* found lower QT interval in participants with normal cognition (n = 224) compared to patients with MCI (n = 77) and Alzheimer's disease (n = 77). These findings came from small cross-sectional studies with patients with cognitive complaints.

Research on ventricular repolarization has been even more sparse. The only study so far that had investigated the association between QT interval and cognitive performance using community-dwelling older adults was cross-sectional and included mostly very old participants (mean = 81, range of 76–85 years).[20] In that study, as in ours, repolarization measurements were not related to cognitive performance.[20] Although additional confirmation is required, our study suggests that subtle changes in the natural rhythm of the heart in midlife are unlikely to affect cognition decades later.

Our study should also be examined in light of its limitations. We did not have information on drugs that can affect the QT interval [43]. However, the low prevalence of use of drugs that may cause QT prolongation in previous studies (2–3%) suggest that our inability to consider drug use will not be a large source of bias.[44] Although we included several factors that could increase the chance of QT interval prolongation, information was missing on other clinical conditions that could influence QT interval (e.g. bundle branch block, hypokalemia and hypocalcemia, endocrine disorders). In addition, the HAAS is a cohort of Japanese American men, and our findings may not hold for other ethnicities and women. Techniques for measuring QT interval have improved since 1965–1971, so it is possible that modern measurements would be more predictive. We also cannot account for any incident prolonged QT interval between Exam 3 and the cognitive measurements. Moreover, we could not examine the association of QT interval with specific cognitive domains since we did not have a complete neuropsychological examination. Although QT interval was not associated with cognitive decline evaluated by CASI, it could be associated with vascular dementia or with cognitive decline in specific domains, such as executive function.

The strengths of this study include a large sample of participants with ECG data and cognitive evaluation over time. In addition, the causal modeling approach used in this study is a strength. MSM with IPEW and IPAW enables robust estimation of the association in contexts where we are concerned about bias due to confounding and attrition. Additionally, we examined for the first time the association of midlife incident prolonged QT interval with later cognitive performance. Since neuropathological lesions associated with dementia may start even two decades before the clinical symptoms of dementia, midlife risk factors are likely most relevant to later cognition.[45] Finally, we present here results from a community-dwelling cohort of adults; thus our results should be generalizable to similar community-dwelling populations.

In conclusion, in a prospective study of midlife ventricular function and cognition in Japanese-American men, we did not find an association of prolonged QT interval in midlife with cognitive performance or decline after 25 years of follow-up. However, future longitudinal studies with different ethnicities and women are important to confirm our findings.

## Supporting information

**S1 Table. Variables included in the denominator of the inverse probability weight models.**
(DOCX)

**S2 Table. Visit-specific estimated inverse probability of attrition and inverse probability of exposure weights.**
(DOCX)

**S3 Table. Mean of final non-truncated weights applied in estimation of the MSM at each visit with cognitive data.**
(DOCX)

**S4 Table. Weighted demographics demonstrating inverse probability weights recover distribution of characteristics at baseline (Exam 2).**
(DOCX)

## Acknowledgments

We would like to thank the staff, scientists, and participants of the HAAS study for making this work possible. We also would like to thank Jess Mez for helping with the design of this study.

## Author Contributions

**Conceptualization:** Claudia K. Suemoto, Laura E. Gibbons, Evan L. Thacker, Lenore Launer, Melinda C. Power.

**Data curation:** Laura E. Gibbons, Lenore Launer, Caroline Phillips, Lon R. White.

**Formal analysis:** Laura E. Gibbons, Evan L. Thacker, Melinda C. Power.

**Funding acquisition:** Lenore Launer, Caroline Phillips, Lon R. White.

**Methodology:** Laura E. Gibbons, Claudia L. Satizabal, Brianne M. Bettcher, Melinda C. Power.

**Project administration:** Lenore Launer, Lon R. White.

**Supervision:** Melinda C. Power.

**Visualization:** Claudia K. Suemoto, Evan L. Thacker, Melinda C. Power.

**Writing – original draft:** Claudia K. Suemoto.

**Writing – review & editing:** Claudia K. Suemoto, Laura E. Gibbons, Evan L. Thacker, Jonathan D. Jackson, Claudia L. Satizabal, Brianne M. Bettcher, Lenore Launer, Caroline Phillips, Lon R. White, Melinda C. Power.

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
