## [Decision Letter · Decision Letter 0]

25 Oct 2019

PONE-D-19-23773

Incident prolonged QT interval in midlife and late-life cognitive performance

PLOS ONE

Dear Dr. Suemoto,

Thank you for submitting your manuscript to PLOS ONE. After careful consideration by 3 Reviewers and an Academic Editor, all of the critiques of all three Reviewers must be addressed in detail in a revision to determine publication status. If you are prepared to undertake the work required, I would be pleased to reconsider my decision, but revision of the original submission without directly addressing the critiques of the three Reviewers does not guarantee acceptance for publication in PLOS ONE. If the authors do not feel that the queries can be addressed, please consider submitting to another publication medium. A revised submission will be sent out for re-review. The authors are urged to have the manuscript given a hard copyedit for syntax and grammar.

**Journal Requirements**

**Comments to the Author**

1. Is the manuscript technically sound, and do the data support the conclusions?

Reviewer #1: Yes

Reviewer #2: Yes

Reviewer #3: Yes

2. Has the statistical analysis been performed appropriately and rigorously? 

Reviewer #1: Yes

Reviewer #2: Yes

Reviewer #3: I Don't Know

3. Have the authors made all data underlying the findings in their manuscript fully available?

Reviewer #1: Yes

Reviewer #2: Yes

Reviewer #3: No

4. Is the manuscript presented in an intelligible fashion and written in standard English?

Reviewer #1: Yes

Reviewer #2: Yes

Reviewer #3: Yes

5. Review Comments to the Author

Reviewer #1: In the manuscript by Suemoto1 et al., it was shown that “Incident prolonged QT interval in midlife and late-life cognitive performance." In this study, the authors tried to show that the time rate of blood pressure variation was a risk factor of developing brain edema. This study is interesting. However, critical flaws are pointed below.

Major comments

#1; Patients with the use of hypertensive therapy.

How many subjects were treated with antihypertensive agent. Several agents might be associated with the study results. Several studies showed that hypertensive status has negative impact on blood brain barrier (BBB) permeability resulting in BBB breakdown. In this point, cerebral autoregulation may be disrupted in the very elderly hypertensive patients. Long acting CCB, ACEI and ARB which does not decrease cerebral blood flow (CBF) are suggested to be appropriate in BP control with high risk at stroke, whereas diuretic which does decrease CBF is not. From these points, it is possible to take the possibility into account that the antihypertensive agents could be affected by the class of antihypertensives agents with cognitive function. How would be the results if the impact of antihypertensive agent class were taken into account in the regression model?

#2: It would be helpful if there was information other than antihypertensive medication (e.g. use of statin, hypoglycemic agent).

#3: Antihypertensive agents after baseline

This question may be similar to that in #1. Althoguh antihypertensive agents that could be used in the baseline would be limited, it would be helpful if there was information about the contents of antihypertensive medication after baseline. These agents change during the follow up might affect the results.

#4: Short QT

Recent notion is not only long QT but also short QT also associated with adverse cardiovascular events. Thus, short QT should be taken into account.

#5: Incidence of Alzheimer disease, vascular dementia or total dementia

I would be interesting if the relationship between QT length and incidence of Alzheimer disease, vascular dementia or total dementia.

Reviewer #2: Paper bySuemoto et all addresses interesting question whether prolonged QT might be associated with decline in cognitive function.

The paper is very clearly written and I do not have any objection to statistical analysis.

However, there are several points which I should raise.

• QT interval is a dynamic parameter affected by several factors, which were not reported or considered in statistical analysis. It should be therefore at least mentioned in limitations of the study the possible effect of presence of bundle branch block, hypokalemia and hypocalcemia, presence of heart failure, ischemia, cerebrovascular disease, endocrine disorders etc.

• There is huge attrition during follow-up. Again, it is usual and inevitable in this kind of study. But the possibility is that it could affect results.

• As QT interval duration affects many drug classes, the estimate that use of this medication is only 2 to 3 % might be underestimated, mainly in older age. Please comment.

• The cut-off value of 370 ms might be too low to be associated with any outcome. Have you considered to use more strict cut-off value, e.g. >400 ms, >440 ms?

Reviewer #3: Dear Colleague

Thank you for this article, which is relevant, and interesting.

The methodology is particularly developed, with particular attention to missing data.

The limits of methodology (and multiple imputation) are detailed in the discussion.

I have some minor comments to submit to the authors:

- Why did not you compared the characteristics of the population according to the extension of the QT in Table 1? I particularly wonder about the difference that there could be concerning the profession (clerical, sales, professional or managerial job) in the 2 groups, and how the authors explain this difference. I think this is a point worth discussing.

- A difference may also be present regarding the ApoE-4 allele (?)

- There are 2 errors in the Total column of Table 1: the total number of "jobs" (708 + 419 = 1127 and not 787), the total number of "hypertension" (435 + 551 = 986 and not 225); the rates of these two results are also incorrect.

Among the limitations, I would add that CASI estimates a cognitive decline (the main cause of which is known to be Alzheimer's disease); perhaps the prolonged QT interval is not a good indicator for cognitive declines in a broad sense, but could be a good indicator of vascular dementia (this is more a perspective than a limit to your work in fact).

Data are not available, but the authors have specified how to access them.

Thank you again for this work,

With kind regards,

Michaël Rochoy, MD, PhD

6. PLOS authors have the option to publish the peer review history of their article (what does this mean?). If published, this will include your full peer review and any attached files.

**Do you want your identity to be public for this peer review?** For information about this choice, including consent withdrawal, please see our Privacy Policy.

Reviewer #1: No

Reviewer #2: Yes: Jitka Seidlerová

Reviewer #3: Yes: Michaël Rochoy

We would appreciate receiving your revised manuscript by February, 2020. To enhance the reproducibility of your results, we recommend that if applicable you deposit your laboratory protocols in protocols.io, where a protocol can be assigned its own identifier (DOI) such that it can be cited independently in the future. For instructions see: http://journals.plos.org/plosone/s/submission-guidelines#loc-laboratory-protocols

We look forward to receiving your revised manuscript.

Kind regards,

Stephen D. Ginsberg, Ph.D.

Section Editor

PLOS ONE

---

## [Author Response · Author response to Decision Letter 0]

21 Jan 2020

Dear Professor Stephen Ginsberg,

We would like to thank you for the opportunity to resubmit the revised version of our manuscript entitled “Incident prolonged QT interval in midlife and late-life cognitive performance” (PONE-D-19-23773) for consideration for publication in PLOS ONE. We are thankful to the referees and the Editor for their thoughtful suggestions that helped to strengthen our manuscript. We have addressed all the reviewers’ concerns, and provided a detail response below. 

Please do not hesitate to contact me if you have any further questions.

We look forward to hearing back from you soon.

Best regards,

Claudia K. Suemoto

 

Reviewer #1

In the manuscript by Suemoto et al., it was shown that “Incident prolonged QT interval in midlife and late-life cognitive performance." In this study, the authors tried to show that the time rate of blood pressure variation was a risk factor of developing brain edema. This study is interesting. However, critical flaws are pointed below.

We investigated the association between prolonged QT interval on cognitive performance, rather than blood pressure variation on brain edema. Therefore, for comments 1 to 3, we did our best to clarify any aspect of the comments that could be related to the association between prolonged QT interval in midlife and cognitive performance in late-life.

Major comments:

1. Patients with the use of hypertensive therapy. How many subjects were treated with antihypertensive agent. Several agents might be associated with the study results. Several studies showed that hypertensive status has negative impact on blood brain barrier (BBB) permeability resulting in BBB breakdown. In this point, cerebral autoregulation may be disrupted in the very elderly hypertensive patients. Long acting CCB, ACEI and ARB which does not decrease cerebral blood flow (CBF) are suggested to be appropriate in BP control with high risk at stroke, whereas diuretic which does decrease CBF is not. From these points, it is possible to take the possibility into account that the antihypertensive agents could be affected by the class of antihypertensives agents with cognitive function. How would be the results if the impact of antihypertensive agent class were taken into account in the regression model?

The role of antihypertensive drugs was not the focus of our study. However, as shown in S1 Table, use of antihypertensive drugs was included in the inverse probability weighting models for having prolonged QT interval as a time-dependent variable, since some of these drugs could influence QT interval duration. Unfortunately, information on specific classes of antihypertensive drugs is not available in the Honolulu-Asia Aging Study. 

2. It would be helpful if there was information other than antihypertensive medication (e.g. use of statin, hypoglycemic agent).

Similarly, we included the use of cholesterol-lowering drugs and hypoglycemic agents as time-dependent variables in the inverse-probability weighting models for having prolonged QT interval as shown in S1 Table.

3. Antihypertensive agents after baseline. 

This question may be similar to that in #1. Althoguh antihypertensive agents that could be used in the baseline would be limited, it would be helpful if there was information about the contents of antihypertensive medication after baseline. These agents change during the follow up might affect the results.

We agree with the reviewer that specific class of antihypertensive drugs could influence QT interval (e.g. angiotensin-converting enzyme inhibitors, angiotensin receptor blockers, beta-blockers, calcium channel blockers). Unfortunately, we do not have the information on the class of antihypertensive medication during the follow-up. Also, the percentage of participants who had hypertension at baseline was small (9%); therefore, this sample is not well-suited for exploring the role of antihypertensive medication classes in the relationship of prolonged QT interval with cognitive decline.

4. Short QT

Recent notion is not only long QT but also short QT also associated with adverse cardiovascular events. Thus, short QT should be taken into account.

Short QT syndrome is a very rare genetic syndrome, which is usually defined as QTc ≤330 ms or QTc interval <360 ms. On the other hand, prolonged QT interval is more common, and it has been consistently associated with stroke, myocardial infarction, and cardiovascular mortality. 1,2 Therefore, the aim of our study is to investigate the association of prolonged QT interval with cognitive performance since prolonged QT is more frequent than short QT in the general population.

5. Incidence of Alzheimer disease, vascular dementia or total dementia

I would be interesting if the relationship between QT length and incidence of Alzheimer disease, vascular dementia or total dementia.

We agree with the reviewer, but this is beyond the scope of this paper. We have added this to the discussion:

“Although QT interval was not associated with cognitive decline evaluated by CASI, it could be associated with vascular dementia or with cognitive decline in specific domains, such as executive function.” (Page 17, Lines 6-8)

Reviewer #2

Paper by Suemoto et all addresses interesting question whether prolonged QT might be associated with decline in cognitive function.

The paper is very clearly written and I do not have any objection to statistical analysis.

Thank you for your comments that helped strengthen our manuscript.

However, there are several points which I should raise.

1. QT interval is a dynamic parameter affected by several factors, which were not reported or considered in statistical analysis. It should be therefore at least mentioned in limitations of the study the possible effect of presence of bundle branch block, hypokalemia and hypocalcemia, presence of heart failure, ischemia, cerebrovascular disease, endocrine disorders etc.

Although we included several factors that could influence QT interval duration, like stroke and myocardial ischemia (S1 Table), we agree with the reviewer that some factors were still missing in the IPEW. We included the lack of information on these factors among our study limitations:

“Although we included several factors that could increase the chance of QT interval prolongation, information was missing on other clinical conditions that could influence QT interval (e.g. bundle branch block, hypokalemia and hypocalcemia, endocrine disorders).” (Page 16, Lines 9-12)

2. There is huge attrition during follow-up. Again, it is usual and inevitable in this kind of study. But the possibility is that it could affect results.

The Honolulu-Asia Aging Study has a long follow-up since individuals were followed from midlife until late life. Therefore, attrition rates are expected to be high. As we too were concerned about the potential for selection bias, we accounted for potentially informative attrition using inverse probability of attrition weights (IPAW). However, we note that this method will only fully mitigate any selection bias present if the assumptions of the method are met; thus, selection bias may not be fully mitigated using this method. We state this limitation in our manuscript:

“Attrition during the follow-up was high in the HAAS. We account for this important problem using IPAW as described. However, some bias due to informative attrition may still be present.” (Page 16, Lines 4-6)

3. As QT interval duration affects many drug classes, the estimate that use of this medication is only 2 to 3 % might be underestimated, mainly in older age. Please comment.

The exact incidence of drug induced QT interval prolongation is unknown, but it is expected to be low.3 In one population-based study showed that about 2–3% of total drug prescriptions may cause unintended QT prolongation in UK and Italy.4 

We discuss the lack of information on drug induced QT prolongation among our study limitations:

“We did not have information on drugs that can affect the QT interval. However, the low prevalence of use of drugs that may cause QT prolongation in previous studies (2-3%) suggest that our inability to consider drug use will not be a large source of bias.” (Page 16, Lines 6-9)

4. The cut-off value of 370 ms might be too low to be associated with any outcome. Have you considered to use more strict cut-off value, e.g. >400 ms, >440 ms?

In fact, we used a more strict cutoff value as suggested by the reviewer. The cutoff of 407 milliseconds was chosen, since it was the 75th percentile of rate-corrected QT interval at Exam 1. Participants were considered to have incident prolonged QT interval if their rate-corrected QT interval was above 407 milliseconds at Exam 2 or 3. 

Reviewer #3

 Dear Colleague

Thank you for this article, which is relevant, and interesting.

The methodology is particularly developed, with particular attention to missing data.

The limits of methodology (and multiple imputation) are detailed in the discussion.

Thank you Dr. Rochoy for your kind comments, which help us to improve our manuscript.

I have some minor comments to submit to the authors:

1. Why did not you compared the characteristics of the population according to the extension of the QT in Table 1? I particularly wonder about the difference that there could be concerning the profession (clerical, sales, professional or managerial job) in the 2 groups, and how the authors explain this difference. I think this is a point worth discussing.

We added P values to Table 1 for univariate tests of differences in characteristics across QT interval groups. Participants with prolonged QT interval were significantly more likely to have clerical, sales, professional, or managerial jobs (34% vs 29%; P = 0.007). We accounted for profession in our analysis as a potential confounder by including it in the inverse probability weight models (see Table S1). Limited evidence in the literature suggests that mental stress (such as might be encountered in some professions) affects QT interval (for example, Andrássy G, et al., Ann Noninvasive Electrocardiol, 2007).5 We chose not to discuss this point in the Discussion because it was peripheral to the purpose of our analysis. 

2. A difference may also be present regarding the ApoE-4 allele (?)

The presence of at least one APOE-4 allele was significantly higher among participants who had prolonged QT interval (21% vs 18%; P = 0.038). We accounted for APOE-4 allele in our analysis as a potential confounder by including it in the inverse probability weight models (see Table S1). 

3. There are 2 errors in the Total column of Table 1: the total number of "jobs" (708 + 419 = 1127 and not 787), the total number of "hypertension" (435 + 551 = 986 and not 225); the rates of these two results are also incorrect.

We corrected the numbers in Table 1.

4. Among the limitations, I would add that CASI estimates a cognitive decline (the main cause of which is known to be Alzheimer's disease); perhaps the prolonged QT interval is not a good indicator for cognitive declines in a broad sense, but could be a good indicator of vascular dementia (this is more a perspective than a limit to your work in fact).

Thank you for your suggestion. We discuss this interesting perspective in the new version of the manuscript:

“Although QT interval was not associated with cognitive decline evaluated by CASI, it could be associated with vascular dementia or with cognitive decline in specific domains, such as executive function.” (Page 16, Lines 18-20)

5. Data are not available, but the authors have specified how to access them.

Yes, data access statement is still valid.

References

1. O'Neal WT, Efird JT, Kamel H, Nazarian S, Alonso A, Heckbert SR, Longstreth WT, Jr., Soliman EZ. The association of the QT interval with atrial fibrillation and stroke: the Multi-Ethnic Study of Atherosclerosis. Clin Res Cardiol 2015;104(9):743-50.

2. Ishikawa J, Ishikawa S, Kario K. Prolonged corrected QT interval is predictive of future stroke events even in subjects without ECG-diagnosed left ventricular hypertrophy. Hypertension 2015;65(3):554-60.

3. Yap YG, Camm AJ. Drug induced QT prolongation and torsades de pointes. Heart (British Cardiac Society) 2003;89(11):1363-1372.

4. De Ponti F, Poluzzi E, Montanaro N, Ferguson J. QTc and psychotropic drugs. Lancet. Vol. 356. England, 2000;75-6.

5. Andrassy G, Szabo A, Ferencz G, Trummer Z, Simon E, Tahy A. Mental stress may induce QT-interval prolongation and T-wave notching. Ann Noninvasive Electrocardiol 2007;12(3):251-9.

---

## [Decision Letter · Decision Letter 1]

10 Feb 2020

Incident prolonged QT interval in midlife and late-life cognitive performance

PONE-D-19-23773R1

Dear Dr. Suemoto,

We are pleased to inform you that your manuscript has been judged scientifically suitable for publication and will be formally accepted for publication once it complies with all outstanding technical requirements.

With kind regards,

Stephen D. Ginsberg, Ph.D.

Section Editor

PLOS ONE

Additional Editor Comments:

1. Please fix the typo as pointed out by Reviewer #1. "First sentence on third paragraph in P16, The phrase "We did not have information" was duplicated."

2. Please fix the terminology as pointed out by Reviewer #2. "Please use same terminology for ECG (EKG vs ECG). In limitations of the study there is text in bold font "We did not have information" which is duplicate."

**Comments to the Author**

1. If the authors have adequately addressed your comments raised in a previous round of review and you feel that this manuscript is now acceptable for publication, you may indicate that here to bypass the “Comments to the Author” section, enter your conflict of interest statement in the “Confidential to Editor” section, and submit your "Accept" recommendation.

Reviewer #1: All comments have been addressed

Reviewer #2: All comments have been addressed

Reviewer #3: All comments have been addressed

2. Is the manuscript technically sound, and do the data support the conclusions?

Reviewer #1: Yes

Reviewer #2: Yes

Reviewer #3: Yes

3. Has the statistical analysis been performed appropriately and rigorously? 

Reviewer #1: Yes

Reviewer #2: Yes

Reviewer #3: Yes

4. Have the authors made all data underlying the findings in their manuscript fully available?

Reviewer #1: Yes

Reviewer #2: Yes

Reviewer #3: Yes

5. Is the manuscript presented in an intelligible fashion and written in standard English?

Reviewer #1: Yes

Reviewer #2: Yes

Reviewer #3: Yes

6. Review Comments to the Author

Reviewer #1: The manuscript is well revised.

Minor comment

First sentence on third paragraph in P16, The phrase "We did not have information" was duplicated.

Reviewer #2: The authors addressed all my comments. I have only minor comments. Please use same terminology for ECG (EKG vs ECG). In limitations of the study there is text in bold font "We did not have information" which is duplicate.

Reviewer #3: Thanks to the authors for their clear and detailed answers. For me, this manuscript is now acceptable for publication.

7. PLOS authors have the option to publish the peer review history of their article (what does this mean?). If published, this will include your full peer review and any attached files.

Reviewer #1: Yes: Michiaki Nagai

Reviewer #2: No

Reviewer #3: Yes: Michaël Rochoy

---

## [Editor Report · Acceptance letter]

11 Feb 2020

PONE-D-19-23773R1 

Incident prolonged QT interval in midlife and late-life cognitive performance 

Dear Dr. Suemoto:

I am pleased to inform you that your manuscript has been deemed suitable for publication in PLOS ONE. Congratulations! Your manuscript is now with our production department. 

With kind regards,

on behalf of

Dr. Stephen D. Ginsberg 

Section Editor

PLOS ONE